# MazeGen: A Low-Code Framework for Bootstrapping Robotic Navigation Scenarios for Smart Manufacturing Contexts

**Ivan Hugo Guevara \*** and **Tiziana Margaria**

Confirm Centre, University of Limerick, V94 C928 Limerick, Ireland
* Correspondence: ivan.guevara@ul.ie

**Abstract:** In this research, we describe the MazeGen framework (as a maze generator), which generates navigation scenarios using Grammatical Evolution for robots or drones to navigate. The maze generator uses evolutionary algorithms to create robotic navigation scenarios with different semantic levels along a scenario profile. Grammatical Evolution is a Machine Learning technique from the Evolutionary Computing branch that uses a BNF grammar to describe the language of the possible scenario universe and a numerical encoding of individual scenarios along that grammar. Through a mapping process, it converts new numerical individuals obtained by operations on the parents' encodings to a new solution by means of grammar. In this context, the grammar describes the scenario elements and some composition rules. We also analyze associated concepts of complexity, understanding complexity as the cost of production of the scenario and skill levels needed to move around the maze. Preliminary results and statistics evidence a low correlation between complexity and the number of obstacles placed, as configurations with more difficult obstacle dispositions were found in the early stages of the evolution process and also when analyzing mazes taking into account their semantic meaning, earlier versions of the experiment not only resulted as too simplistic for the Smart Manufacturing domain, but also lacked correlation with possible real-world scenarios, as was evidenced in our experiments, where the most semantic meaning results had the lowest fitness score. They also show the emerging technology status of this approach, as we still need to find out how to reliably find solvable scenarios and characterize those belonging to the same class of equivalence. Despite being an emerging technology, MazeGen allows users to simplify the process of building configurations for smart manufacturing environments, by making it faster, more efficient, and reproducible, and it also puts the non-expert programmer in the center of the development process, as little boilerplate code is needed.

**Keywords:** smart manufacturing; machine learning; evolutionary computing; navigation scenarios

## 1. Introduction

Robotic Navigation Scenarios are configurations with obstacles and objects, such as walls, tables, machines, production lines, etc., in a typically delimited or enclosed space, that enable us to test different types of requirements over a spatial model. This model approach gives the possibility to find a cost-effective solution and to reach preliminary conclusions for decision-making on two relevant categories of problems: how to efficiently navigate a given space, and how to design a space, e.g., by appropriately placing objects, in such a way that effective and efficient navigation is possible. Several examples can be found in research with interesting setups in robotics. In [1], the authors successfully place "virtual obstacles" in the navigation scenario, which is linked to the real-world robotic scenario, to prevent the robot from taking a tentative route and force it instead to looking for an alternative route, in order to guide it to a safer path. In [2], another solution for a simulated environment to test robotic models foresees a high-fidelity 3D simulator that models real-world crowd behavior, sensor noise, frictions, and delays of the robot, to close the sim-to-real gap. There are "hybrid approaches", where the real-world setup is combined

with the simulated one, as in [3]: to test an agricultural autonomous robot system, different images were stitched together into a 20 m-long strip, which tried to replicate rows of the crop, and were used as a texture added to the ground plane in the simulator. In this context, the advent of more powerful machine learning models opens new opportunities within the Smart Manufacturing field: it enables finding optimal unseen solutions for robotic test navigation controllers on smart agents [4], bringing contextual intelligence to supply chains by minimizing running costs to successfully manage inventory [5], or finding the most efficient disposition using elements such as obstacles, middle/corners walls and the goal involved in the landscape [6].

There are also some interesting use cases such as [7], developing a framework for steering fault diagnosis, a hybrid model using a model-based residual generator, and a Support Vector Machine algorithm. The model also has the possibility to deal with undersampled data, employing a LDA method (linear discriminant analysis) to form a balanced training dataset and an optimization method called Grey Wold Optimizer (GWO), which improves the classification accuracy. The proposed method obtained a higher G-mean than the ones derived with a benchmark in eight tests out of seventeen datasets and was also applied to fault diagnosis of the field data for a vehicle steering actuator, also outperforming benchmark methods for all three kinds of faults. Additionally, [8], using evolutionary algorithms, an optimization approach based on the PSO algorithm, investigated a tuning problem of PID controller parameters for a CAN-based DC-motor system. The corresponding upper bound is obtained by analyzing the CAN-induced delay, in order to tune the parameters of the PID controller, and we can also cite [9], using a novel approach for battery EOL (end-of-life) prediction. The novel approach uses a KF (Kalman filter), which applies to the available partial battery degradation data, without depending on the empirical degradation model and only on the virtual degradation rate and acceleration. Then, the prediction is executed through an iterative GPR (Gaussian Process Regression) with a moving sliding window. The results show the novel approach outperforms the traditional one from two perspectives: the predicted value is closer to the true one and the proposed method has a smaller range of prediction uncertainty.

In our particular use case, we use a machine learning technique from the branch of Evolutionary Computing called Grammatical Evolution (GE) [10–12] that uses elements from Evolutionary Algorithms to provide novel and original solutions. GE uses the following approach to find the optimal solution according to specified criteria: It begins by creating a set of random individuals, called a population, that is described by integer arrays. The set of rules for the structure of the solutions are described by a Backus–Naur Form grammar (BNF) [13]. The algorithm converts the initial population (the integer arrays) into new solutions through a mapping process that uses the BNF grammar and the modulo MOD operator. The final result is a new population where different solutions are derived from the grammar and thus conform to its rules. In order to find the optimal solution, each population needs to go through an evolutionary process where genetic operators (mutation, crossover, selection) are applied to some of the individuals, and some individuals are discarded. This way, more genetic variation is brought among the individuals of successive population generations. A fitness function guides the search for better solutions by defining a metric to measure each individual. This evaluation also produces metadata about the evolutionary process, allowing us to see whether the fitness improves or becomes worse. GE has been successfully applied to different domains, e.g., design, architecture, and engineering [14], where individuals are synthesized to form different engineering design solutions, search-based software engineering [15] optimizing run-time performance in the regular expression language, program synthesis [16] automatically generating caching algorithms, sports analytics [17] predicting matches for the Six Nations Rugby, animation [18] to different simulated animals, design of a cryptographically secure pseudo-random number generator [19], the evolution of complex digital circuits with SystemVerilog [20] and optimizing combinational logic circuits [21]. Given these successful use cases, we aim to deliver a novel solution for automatically generating maze scenarios

that can be used in the Smart Manufacturing environment and can help us to define the best disposition of things within a space, in order to have an efficient and cost-effective navigation solution.

To accomplish this, we developed a tool called PyGEVO [22], capable of empowering non-expert people and building powerful experiments with little programming experience, having sufficient high-level abstractions instead of boilerplate code to rely on and, also, is already tested, documented and ready to be used. Regular programming tools solve difficult problems and converge to possible solutions, but there still exist some key issues required to be addressed, such as (1) the development of heterogeneous architectures, where researchers might deal with highly-complex software architectures, making them prone to bugs and errors. (2) Good programming skills are needed to build the software ecosystems and (3) edge-cutting technologies tend to have little support, which may lead to unexpected behaviors in the system. We believe low-code programming is a great approach for anyone to easily bootstrap difficult tasks with little effort, that is why the contributions and highlights of this article can be summarized as follows:

1. Design and development of a machine learning low-code tool (PyGEVO) to bootstrap experiments in a more straightforward way.
2. A novel approach to automatically generate navigation mazes through evolutionary algorithms, allowing customized restrictions and the selection of different criteria.
3. Integration with a 2D graphical framework (Kivy) to visualize the state of the mazes.

## 2. Materials and Methods

### 2.1. Choosing Our Figures

We start by defining the building blocks that will compose our mazes: an angle, a square, an E-shape, and an L-shape, which are simple geometric shapes acting as obstacles for the navigation robot. Although there are no restrictions in terms of shape type, as it is quite straightforward to create them within the Kivy framework, we chose that set of figures because it was easy to represent them: an angle requires two lines with the same starting point but different ending positions, the square only four lines aligned, the e-shape three parallel lines and one crossing vertically and the L-shape two lines with the same disposition as the square. All these figures were made with the Kivy framework, by defining the disposition of the lines and concatenating them to create the corresponding figure, if needed.

### 2.2. Describing the Maze Components: The BNF Grammar

Having defined the figures composing our mazes, we added them into the BNF grammar and additionally, we allow composed figures, where a composed figure has a (possibly infinite) set of simple obstacles. Its number is limited in practice by the hyperparameters we indicate for the experiment. The Grammatical Evolution approach uses grammar in BNF form to describe the possible alternatives in terms of production. Figure 1 shows two main rules: one for the <scenario> patterns, that foresees in this specific example five different alternatives, and one for <composedFigures>, which consist of four different simple obstacles followed by a <scenario> again. The two productions are in fact mutually recursive. This grammar describes in a synthetic way all the possible sets of obstacles a scenario may contain, which are the language it generates. BNF grammars are in fact the standard format to describe programming languages in Computer Science: from simple to complex ones such as Java and C, they have a BNF description. In this context, the entire space of solutions will be defined by the different derivation rules, describing a tentative solution space. The meta-operator OR separates patterns, here written as |, and the alternatives can be enumerated, e.g., from left to right, and we use here the enumeration index as an integer that plays a role in the encoding of the individuals and mapping of the rules to solutions. BNF grammars can be recursive, and in this case, we see that each production has one or more patterns that "call" the other production. The BNF, which is a core tool for the construction of compilers for programming languages, is also a validator of the

syntactic correctness of the programs, as each correct program must be generable along the grammar of its programming language. BNFs are thus excellent representations to do both structure-guided synthesis and structure-guided validation/testing.

**Figure 1.** Evolutionary process for GE.

*2.3. Description of the Evolution Process*

In GE terms, a maze is a tentative solution described by its genotype, consisting of an array of integers. Each integer in the array is called a codon, in analogy to the terminology in use for genetic information in biology. The genotype of an individual of the population encodes, codon by codon, the sequence of pattern choices in the layered BNF that describe that specific individual. Given a genotype, the individual it represents is generated by successively mapping each codon to the BNF. Starting from the left, the leftmost codon is associated with the top rule of the BNF, and the specific pattern in that rule is identified by applying to the integer in the codon the modulo operator $MOD_n$, where $n$ is the number of patterns in that rule. Next, the leftmost Non-Terminals in that pattern are associated with the next codon: the corresponding BNF rule is now examined, and the value of the codon $MOD_n$, this rule's $n$, again determines the chosen pattern for further substitution. If the pattern has no further Non-Terminals, the substitution terminates and moves to the next non-terminal of that pattern or higher-level patterns that still need substitutions.

Once an individual is fully generated, a fitness function is applied to it, to determine how good it is in the context of the current population. The fitness level of each individual within the current set of individuals determines which of them are relatively more "apt to survive", therefore the best individuals to be directly included in the next generation, or used as "parents" to produce offspring through genetic recombination operations of their genotype.

The evolution process is illustrated in Figure 2 with more detail. The genotype of the initial population is randomly generated. Each generation starts with an initial population, here containing individuals $P_0 \ldots P_n$, each described by their genotype. In this example, the genotype has length 8 codons, represented as an array of integers of length 8, where $P_0$'s genotype is [34 10 200 80 5 70 45 43]. For each individual, as described, the leftmost codon is associated with the first production in the grammar (here, <scenario>), which has 5 alternative patterns. Applying the respective MOD 5 operator to the integer value of the codon determines which alternative to choose. For the individual $P_0$, its first codon, 34, refers to the production <scenario>, which has 5 alternative patterns, thus 34 MOD 5 is computed and the 5th option <composedFigures> is chosen (As MOD counting starts with 0, the output of a MOD 5 operation is in the range [0...4].). Then, 34 MOD 5 = 4, so the 5th pattern in the rule <scenario> is chosen, which is <composedFigures>: so it has been determined that the individual corresponding to $P_0$ is a composed figure. Analogously, the next codon of $P_0$, 10, is now evaluated with regard to the production <composedFigures>, which has 4 alternatives. Thus, 10 MOD 4 = 2, corresponding to the 3rd pattern e-shape (<scenario>), so $P_0$ contains an E-shape obstacle and the recursion in the grammar brings the evaluation of the next codon according to the <scenario> production. The entire scenario is derived in a similar fashion until reaching a terminal (i.e., a simple shape in the <scenario>). There is no more derivation and the rest of the genotype is therefore irrelevant. In this example, $P_0$ is a scenario containing an E-shape and an angle.

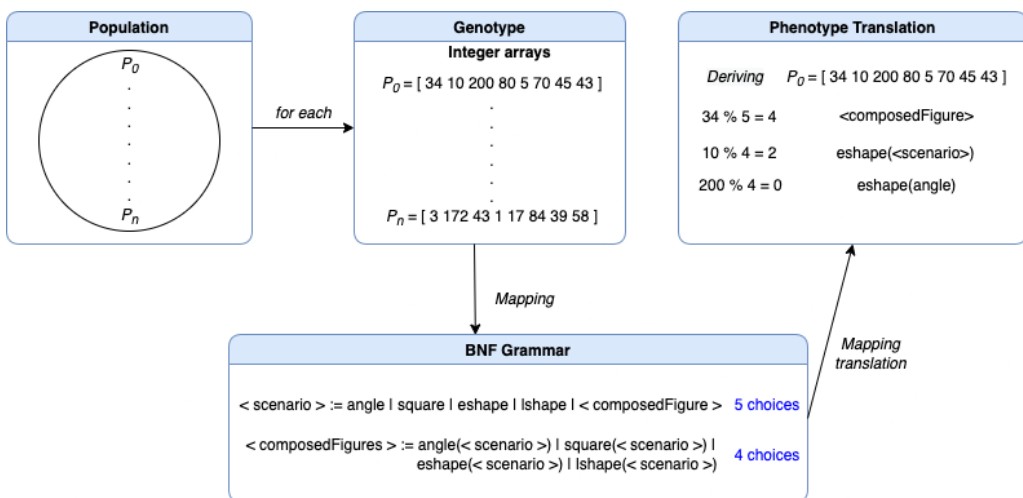

**Figure 2.** Evolutionary process for GE: from the population to the individuals.

This population computation phase is followed by the evolution step, where genetic operations are applied to the genome of the individuals selected to produce the next generation, in the hope to achieve better and improved individuals. The evolutionary process proceeds generation by generation, identifying in the current generation its best-performing solutions according to the fitness function, and then producing a new generation by applying the evolutionary algorithm. The algorithm will keep in the successive generation only a percentage of the currently most apt individuals (selection), and it will also mutate and recombine various individuals according to recombination rules. From the population perspective, evolution will guide this way the most efficient solutions toward the production of the final generation, which should be the fittest. The definition of fitness is clearly central to the selection, as it encodes the notion of preference for individuals with certain characteristics.

### 2.4. Defining the Experiments

To test the different grammar and fitness functions and guide the evolutionary process, PyGEVO, the low-code novel Grammatical Evolution framework, is used. We carried out 8 experiments with different configurations as reported in Table 1. In the first place, we defined 2 different versions of the grammar, as we can see in Figures 3 and 4. The first will include an earlier approach, where the hyperparameters from the elements were placed in the grammar itself, relying on the different terminals in the file. The second version of our BNF file had a performance improvement, by placing the hyperparameters in our new software architecture. This way PyGEVO had less I/O processing as all the objects are stored in RAM memory or the cache, greatly improving the experiment processing.

```
<scenario> :=  triangle(<size>,<number>,<number>) | square(<size>,<number>,<number>) | eshape(<size>,<number>,<number>,<number>,<number>)
|        |        |  lshape(<size>,<number>,<number>) | <composedRules>
<composedRules> := triangle(<size>,<number>,<number>,<scenario>)  | square(<size>,<number>,<number>,<scenario>)
|        |        |        | eshape(<size>,<number>,<number>,<number>,<number>,<scenario>)  | lshape(<size>,<number>,<number>,<scenario>)
<size> := 1 | 2 | 3 | 4 | 5 | 6 | 7 | 8
<number> := 100 | 200 | 300 | 400 | 500 | 600 | 700 | 800
```

**Figure 3.** First version of the grammar.

```
<scenario> :=  triangle() | square() | eshape() | lshape() | triangle(<scenario>)  | square(<scenario>)  | eshape(<scenario>)  | lshape(<scenario>)
```

**Figure 4.** Second version of the grammar.

**Table 1.** Experiments set up: hyperparameters and configurations for the experiments.

| Hyperparameters | Experiment 1 | Experiment 2 | Experiment 3 | Experiment 4 | Experiment 5 | Experiment 6 | Experiment 7 | Experiment 8 |
|---|---|---|---|---|---|---|---|---|
| Runs | 15 | 15 | 15 | 15 | 15 | 15 | 15 | 15 |
| Individuals | 1000 | 5000 | 1000 | 5000 | 1000 | 5000 | 10,000 | 50,000 |
| Genotype Size | 32 | 32 | 32 | 32 | 32 | 32 | 32 | 32 |
| Generations | 15 | 15 | 15 | 15 | 15 | 15 | 30 | 30 |
| P. Selection | 0.1 | 0.1 | 0.1 | 0.1 | 0.1 | 0.1 | 0.1 | 0.1 |
| F. Function | Higher→ better | Higher → better | Higher → better | Higher → better | Penalize | Penalize | Penalize | Penalize |
| BNF Version | v1.0 | v1.0 | v1.5 | v1.5 | v1.5 | v1.5 | v1.5 | v1.5 |

Each experiment was run 15 times in order to obtain a set of tentative solutions (in terms of the different shapes that could fit the given scenario) and results (time spent running the experiments, consumed resources, failure, success rate, etc.), and to achieve this way conclusions for that specific experiment. The overall population size was between 1000 and 5000 individuals in the first six examples, with a genotype of length 32, i.e., 32 integers. Those solutions evolved through 15 generations with a selection of 10% individuals from generation to generation. In this context, we cloned the selected individuals and applied a genetic operation (mutation) to one subset and crossover to the other one, concatenating both to select the aptest.

In the last two experiments, a larger population was used (10,000 individuals for experiment 7 and 50,000 for experiment 8) and also a longer evolutionary time frame. We added more generations (a total of 30) to see the impact it would have on the quality of the outcomes.

We also evaluated two fitness functions:

- The first one (Experiments 1 to 4) uses a simple criterion: the higher the number of objects in the scenario, the better the scenario is. This criterion, however, does not consider collisions between obstacles: some scenarios contained overlapping objects, leading to unrealistic/unfeasible scenarios.
- The second fitness function, used in Experiments 5 to 8, adopts a hybrid approach, penalizing objects that tend to be bigger, and thus are more likely to collide with the rest. In these experiments, we considered invalid any solution with an element size over 500 pixels.

### 2.5. Defining the Software Architecture

In order to speed up the evolutionary process, we simplified the grammar by calculating the size and position of each element in a new class (RandomGenerator) and also added a cache to look for repeated individuals. As we already mentioned, if the size and position were part of the grammar definition, the algorithm would need more nodes to recreate the production, making the experiment more costly in terms of computing resources. As a consequence, many more nodes would be needed for each element in the maze, requiring a much longer genotype, and also needing more I/O operations, therefore making the process slower.

Instead of relying on the BNF for describing and determining these features, our software architecture, shown in Figure 5, deals with this in a second step. The architecture of the experimental setup uses PyGEVO to manage the BNF and the genotype-to-phenotype mapping, and Kivy [23], a free and open-source Python-based graphical framework, to visualize the scenarios. Both tools are available as open source and provide good documentation for development.

Specifically, the main orchestration occurs in a Python script, which defines the fitness function and invokes PyGEVO. PyGEVO abstracts all the logic for the evolutionary process and the synthesis of the phenotypes, handling the grammar-related part of the GE approach. In this approach, a connection with a random number generator provides the number and size generation. This simplifies the BNF grammar, which describes only the scenario structure, but not the attributes. The cache helps us to achieve better performance by managing repeated individuals. Considering that through the entire process, random populations are generated from a relatively short genotype, it is likely that the same genotype may occur

multiple times. To avoid recomputations, the cache stores the genotype/phenotype pairs that have been already produced, and the successive times the precomputed phenotype is directly retrieved. Finally, a Scenario Builder bridges the phenotypes coming from PyGEVO and the Kivy framework, providing a translation into the visualization language supported by the graphical library. As soon as phenotypes are derived, they are translated into Kivy code and placed into a proper log. To this aim, we created a simple *Domain Specific Language* for the *ScenarioBuilder* interface, to easily handle the creational part of the elements.

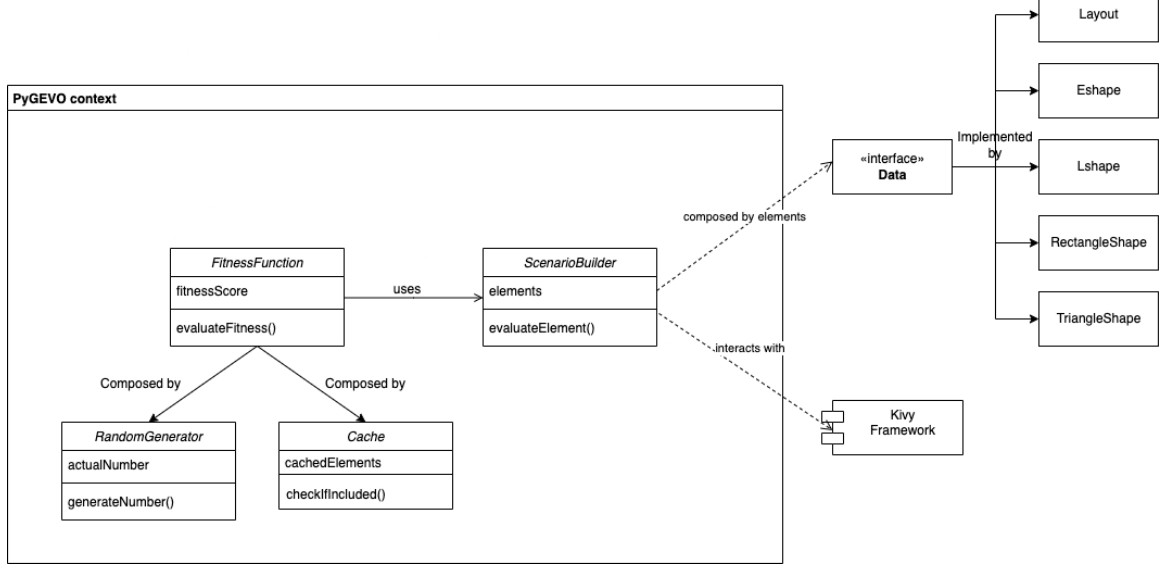

**Figure 5.** Software architecture: PyGEVO (**left**) and Kivy (**right**).

### 2.6. Defining the PyGEVO Code

As we already mentioned, PyGEVO exhibits an easy and straightforward way to define experiments in a few lines of code. As we can evidence in Figure 6, four lines of code are enough to create and execute the experiment (except the optional lines for measuring time). To accomplish this, some necessary steps are needed:

1. Install PyGEVO: we can install the python simply by typing *pip install pygevo*.
2. Define the grammar file: establishing the universe of solutions available for the evolutionary process.
3. Define a fitness function: a piece of Python code defining the criteria to evaluate in each case, or also use a built-in one.
4. Define the hyperparameters: In this case, we defined *numberIndividuals* (as the number of random individuals to create), *individualSize* (as the number of codons for each individual), *generations* (number of times the population will be processed), *porcentSelection* (percentage of individuals will be selected to perform genetic operations), *fileSave* (whether we want to save the result in a file or not) and order established to show the individuals (sorted or not, *reverse*).
5. Create the necessary python objects (*pop*, *population* and *algo*).
6. Run the experiment using the *evolveWithGE_v1()* method. All the individuals will be contained in a collection called *evolvedPop*.

The result will contain the evolved solutions with three properties populated: *fitnessScore* (the score given by the fitness function), *phenotype* (the solution in terms of the BNF), and *genotype* (the "DNA" from the individual, represented by an array). In order to simplify the workflow, we will be able to find the maze structure in the *phenotype* property in Kivy format. This way we will be able to execute and see the 2D maze generated by the ML agent using the Kivy framework, just by passing the parameter we need and visualizing our solutions.

```
import Examples.MazeGen_Generator.fitnessFunctionv2 as ff
from core.domain.population import Population
from core.domain.algorithms import Algorithms
import time

start = time.process_time()
pop = Population("grammar-v2.bnf", numberIndividuals=1000, individualSize=32,  fitness_function=ff.fitnessFunction)
population = pop.generatePop()
algo = Algorithms("grammar-v2.bnf", initBNF=0, debug=False)
evolvedPop = algo.evolveWithGE_v1(population, pop, gen=10, porcentSelect=0.6, fileSave="",reverse=True)
print(time.process_time() - start)
```

**Figure 6.** Piece of code using PyGEVO.

## 3. Results

This section shows the results of the described experiments, using a subset of the mazes generated by the ML agent.

### 3.1. Solutions from the ML Agent

The solutions provided by the ML agent during the evolutionary process are very diverse. The produced mazes consist of several types of figures with different shapes, widths, and sizes. The shapes consist mostly of lines, angles, L-shapes and e-shapes displayed in a scenario, greatly differing in characteristics. As a first approach for the PyGEVO-based ML agent, and despite showing randomness and a lack of semantic meaning in the position of each figure, the collection of outcomes shows that this is a good first approach for the framework, enabling further improvement in the next versions.

Figures 7–10 show scenarios 1 to 4 and we can appreciate the following configurations:

- Scenario 1 consists in 16 elements (7 l-shapes, 1 angles, 8 lines).
- Scenario 2 consists in 16 elements (3 l-shapes, 2 angles, 7 e-shapes, 4 squares).
- Scenario 3 consists in 14 elements (3 squares, 4 l-shapes, 3 angles, 4 e-shapes).
- Scenario 4 consists in 12 elements (3 l-shapes, 3 angles, 3 squares, 3 e-shapes).

The same ones are driven by a configuration with low restrictions. In the next pages, we will describe the decision-making process to improve the scenarios, as well as the performance and semantic meaning.

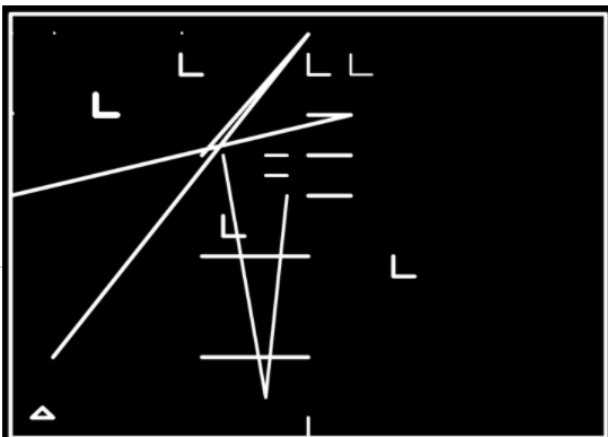

**Figure 7.** Scenario 1–16 elements.

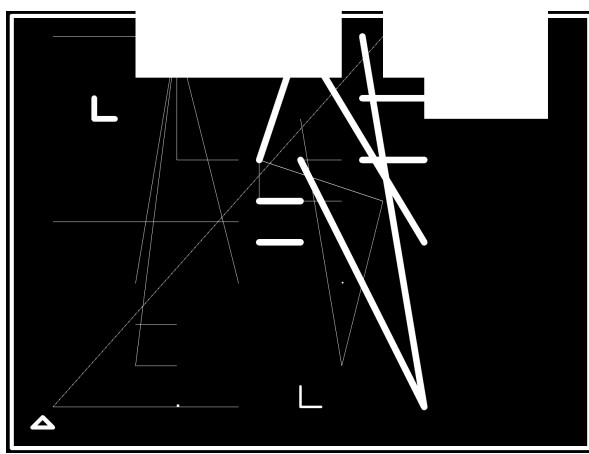

**Figure 8.** Scenario 2–16 elements.

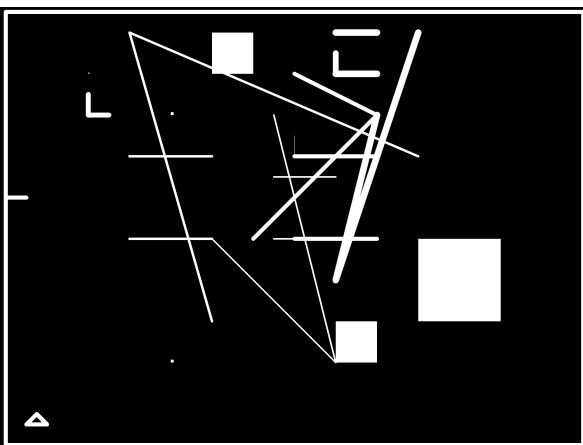

**Figure 9.** Scenario 3–14 elements.

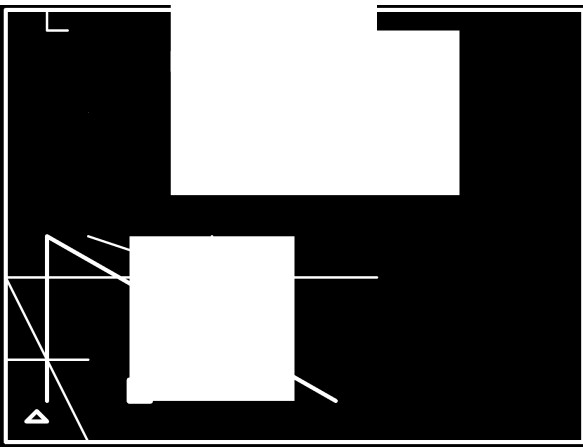

**Figure 10.** Scenario 4–12 elements.

*3.2. Performance Measurement from the ML Agent*

The experiments using the first version of the BNF (numbers 1 and 2) were the ones taking most of the time: eight days for the first using 1000 individuals and twelve days for the one using 5000. The performance increased drastically once we moved to the next grammar version, taking two days in total for experiments 3 to 8. Empiric evidence showed no big difference in terms of performance between BNF v1.5 without penalization vs. BNF v1.5 with the fitness function penalizing big objects (greater than 500 px), so we decided to show them as one. In total, we spent twenty-two days running all eight

experiments (Datasets and graphics can be found in: https://github.com/IvanHGuevara/Results-MDPI-MazeGen (accessed on 11 April 2023)), as detailed in Figure 11.

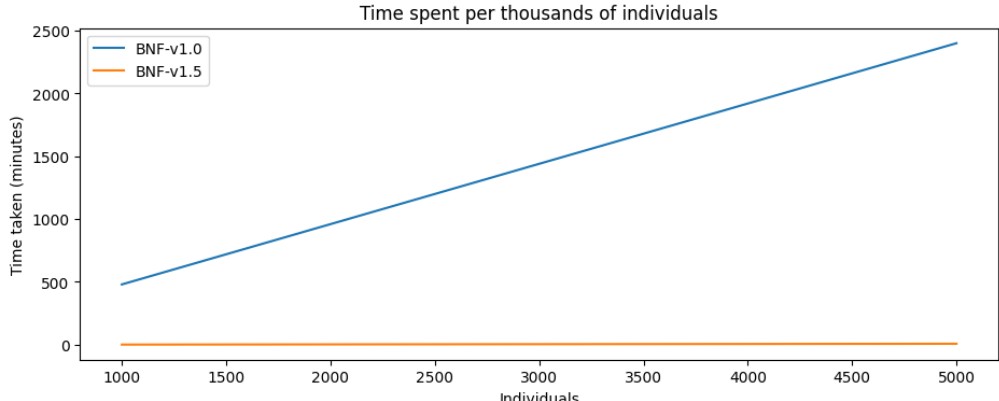

**Figure 11.** Time spent by experiment—Comparison of the different BNF grammars.

To analyze the fitness score evolution, as we can see in Figure 12, we took a subset of solutions of the 15 runs from each experiment, a total number of 93 individuals for each one, and generated a mean fitness score to have a significant metric. In the case of the first version, the mean fitness score obtained was 4, the worst scenario presented due to the quality of the mazes and the time spent developing them. The second grammar version (v1.5) improved, not only by reducing the time of the evolutionary process but also increased the mean fitness score up to 12. Despite this result, and as we already mentioned, the second grammar file also had issues by allowing big objects (more than 500 pixels) to appear in the scenarios, which is why we created a different fitness function that penalized those objects. We managed to reduce the appearance of these ones, but at the same time, the mean fitness score was reduced to almost 10.

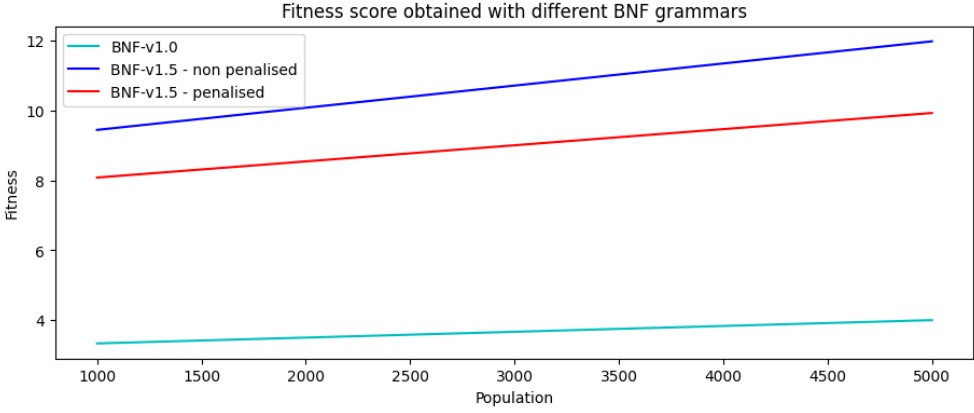

**Figure 12.** Fitness function projection depending on the grammar version.

*3.3. Limitations of MazeGen 1.0*

The work done with MazeGen in its first version produced important results: it is a novel approach for recreating smart manufacturing environments, significant performance improvements were achieved by caching repeated solutions, and the developed software architecture succeeded to improve the grammar mappings and to integrate a graphical framework.

However, there are issues that need to be addressed in order to use it as a real-world 2D tool:

- The scenarios are abstract, they lack a useful connection with real-world scenarios,

- The use of simplistic geometric figures to represent obstacles is not directly related to the smart manufacturing ecosystem, although walls, benches, bounding boxes, and angles of incidence for trajectories connect well with the shapes we chose,
- The fitness function has a simplistic criterion (the higher the number of elements, the better the maze), which is useful to test performance, but even in the modified form of Experiments 5 to 8 is still raw.

In order to partially address these limitations, we are developing a second version of the system: MazeGen 2.0.

## 4. Improving Delivery of Smart Manufacturing Scenarios: MazeGen 2.0

In MazeGen 2.0, we changed the grammar and the figures in the respective scenarios. The initial grammar had a triangle, a square, an e-shape, and an l-shape as basic forms, and included recursion to allow scenarios with possibly infinite recursive figures. The second grammar, shown in Figure 13, describes more directly the smart manufacturing application domain addressed in the Confirm Research Centre. The scenario elements are now a cobot (Figure 14), a router (Figure 15), a vertical wall (Figure 16), a production line (Figure 17), a horizontal wall (Figure 18) and the recursive forms of each. They are still represented in the Kivy graphical framework, in order to simplify the development of the figures, but we are considering moving to sprites in order to also allow a dynamic movement of the figures.

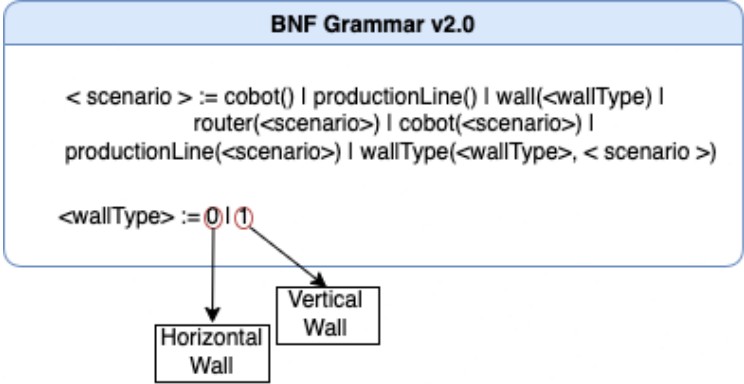

**Figure 13.** New grammar defining realistic environment.

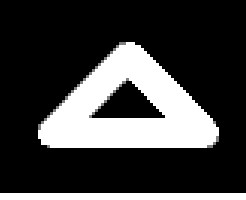

**Figure 14.** Triangle representing the cobot.

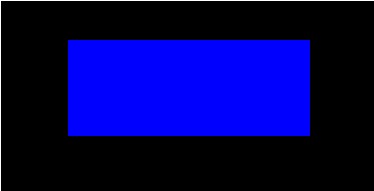

**Figure 15.** Blue square representing the router.

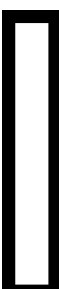

**Figure 16.** Vertical wall.

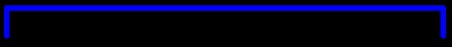

**Figure 17.** Production line represented in blue.

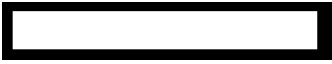

**Figure 18.** Horizontal wall.

### 4.1. A New Fitness Function: Penalizing Overlapped Objects

We modified the criteria for the fitness function: this third version will take into account overlapping figures. As mentioned in Section 1, the first version of the fitness function (Equation (1)) considered the sum, only of the number of times that figure appeared (indicated by $f_t(Fig_i)$), trying to generate figure-rich scenarios: the more figures the maze had, the better. It was not aware of overlapping geometric figures, which is an essential trait for producing realistic scenarios. We improved that fitness function (Equation (2)) by considering the number of overlapping elements ($N_{overlap}$) and subtracting from the total number the number of overlaps multiplied by a coefficient ($K_{fig}$), that is right now 0.5. The intention is to heavily penalize the occurrence of overlapping figures. By choosing ($K_{fig}$) = 0.5, no scenarios with overlaps can be better than those without. Figure 19 illustrates this property: the two scenarios shown to have the same number of figures, but the left one overlaps and the right one does not. With the first fitness function, their fitness score ($G_t$) is 4 for both scenarios. With the new definition of fitness, the non-overlapping scenario has still a fitness score of 4, but the fitness of the scenario with overlaps drops to 2.5.

$$G_t = \sum_{i=1}^{n} f_t(Fig_i) \quad \text{where } \forall n, i \in \mathbb{N} \tag{1}$$

$$G'_t = \sum_{i=1}^{n} f_t(Fig_i) - (K_{fig} * N_{overlapped}) \quad \text{where } \forall n, i \in \mathbb{N} \tag{2}$$

### 4.2. Preliminary Results for MazeGen 2.0

MazeGen 2.0 is still an ongoing work and under development, but we have already obtained preliminary results and some insights about the new approach taken into consideration. As shown in Figures 20 and 21, respectively, metrics did not improve with the new configuration as the time spent is higher than BNF v1.5 and the fitness score in the same range as the first version of the grammar, meaning a significant score decrease with fewer possibilities of having more figures in the scenario. Both decisions are realistic for the kind of use cases we are concerned with, but as seen in Figures 22–25, MazeGen 2.0 requires some work to produce scenarios with more figures and intersections. This is mainly caused by two major changes: (1) the use of the new fitness function that penalizes, not only big figures (greater than 500 px), but individuals with overlapping figures, and (2) carrying out experiments with an individual size of 16, which naturally generates fewer figures.

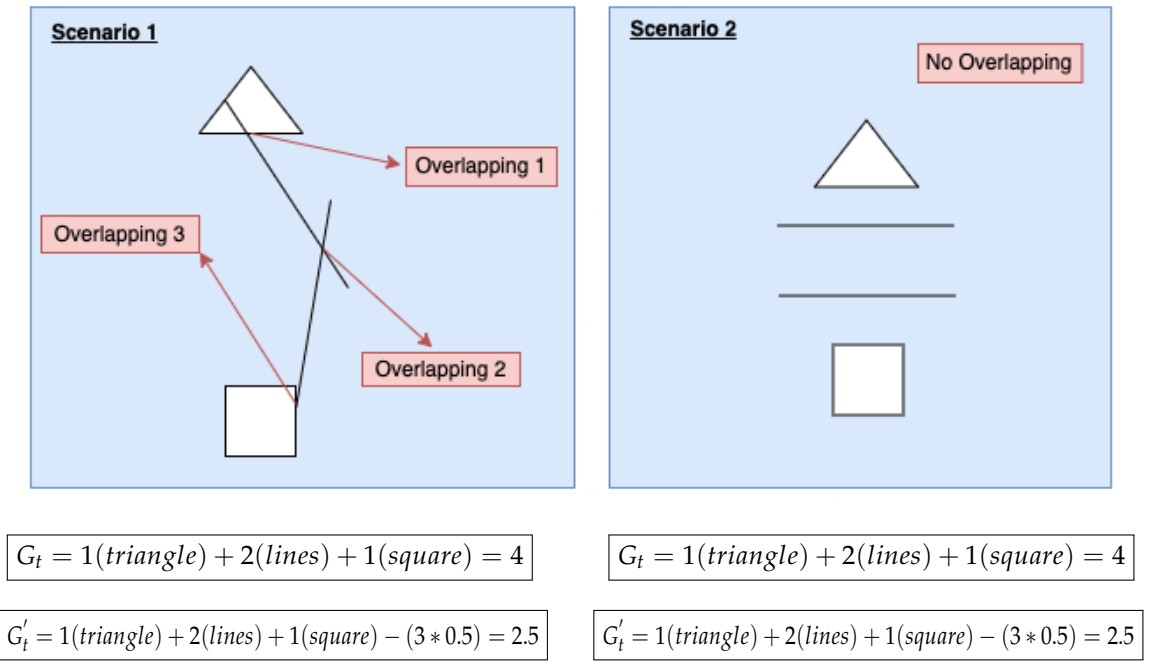

**Figure 19.** An overview of both approaches with the different fitness functions is presented.

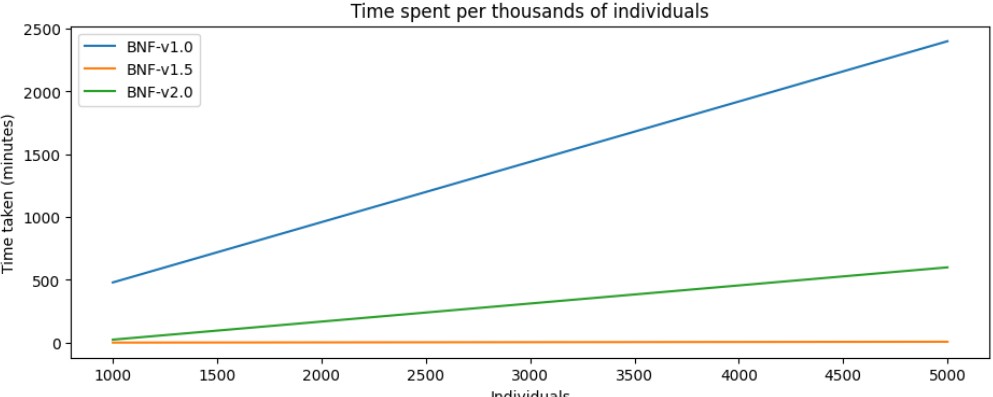

**Figure 20.** Time spent metric taking into consideration the new grammar version.

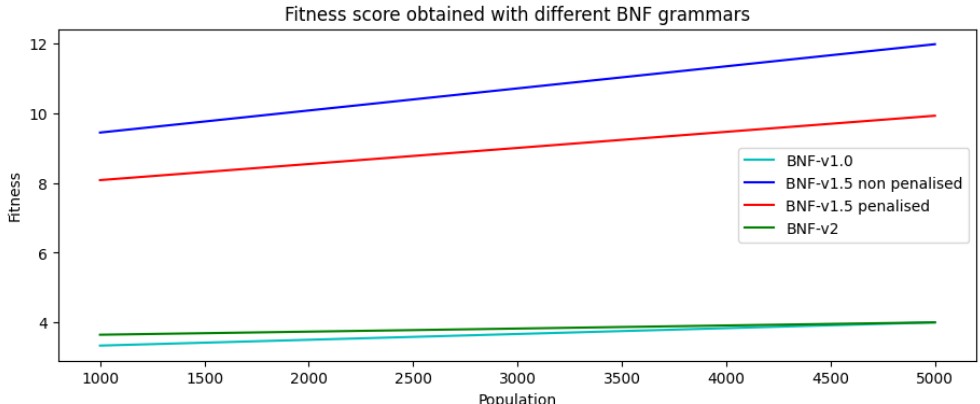

**Figure 21.** Fitness per population metric taking into consideration the new grammar version.

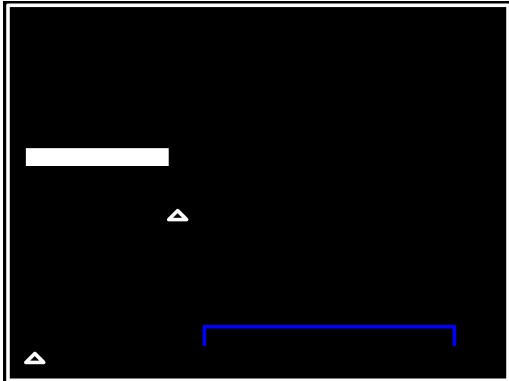

**Figure 22.** Scenario 1–4 elements.

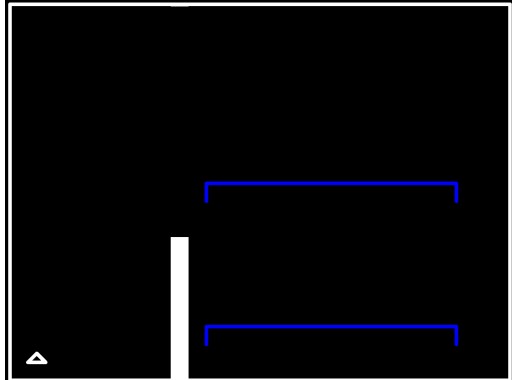

**Figure 23.** Scenario 2–4 elements.

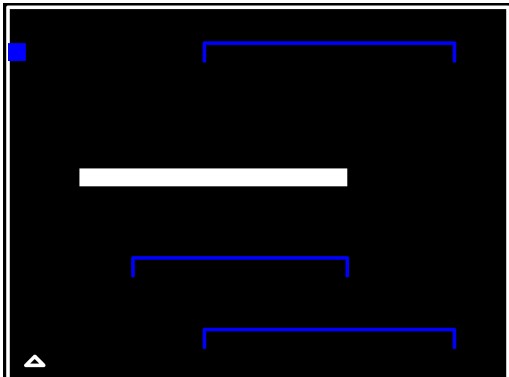

**Figure 24.** Scenario 3–6 elements.

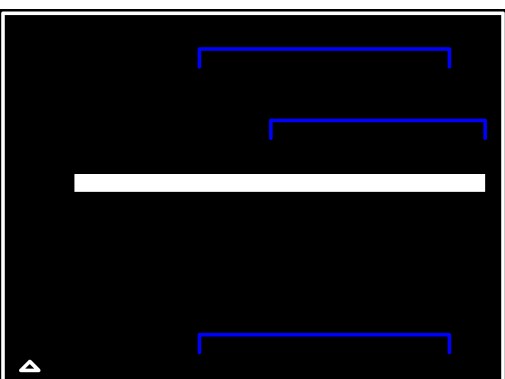

**Figure 25.** Scenario 4–5 elements.

A central question is to determine the size of the initial population needed in order to provide good solutions. Penalizing the overlapping individuals causes a good part of the generated population to be filtered out, thus progressing fewer individuals to the next generation. For this reason, we need to consider several strategies: a larger initial population starting with more individuals helping to compensate for the larger loss while going through the evolutionary process or injecting during the evolutionary process itself a random set of individuals to see whether or not it improves the score.

Another question concerns how to best handle the fact that the penalizing fitness function tends to lead to solutions whose elements occupy less space. Scenarios with big elements (such as the production line or the walls) tend to have overlapped, so individuals with smaller figures are more likely to survive. This effect decreased the shape and size variety, and therefore the semantic coverage of the overall solution. This discovery is leading us to rethink again the fitness function, in order to deliver more balanced solutions in terms of the number of figures and the diversity of elements. The current fitness function has improved, but still does not satisfy real-world use cases for maze generation. In fact, as of today, the ML agent is not yet capable of figuring out which maze, in a smart manufacturing environment, is a more cost-effective solution than another. By cost-effective we mean that, given 2 scenarios, the more cost-effective has the most efficient configuration if we want to implement it in a real-world use case. The cost-effectiveness question is relevant in practical cases: for example, given a maze with 15 production lines and another one with 5, which one is most cost-effective in terms of geometric figures in order to produce a certain amount of assets?

## 5. Conclusions

The automatic generation of maze scenarios through Evolutionary ML is a novel way to face the construction of smart manufacturing mazes, providing variety in terms of geometric figures and entirely delegating to the algorithm the design and semantic sense of each building block disposition. We can summarize our achievements as follows:

1.  Successfully created a flexible low-code framework (PyGEVO) to deliver an easy way to create difficult evolutionary computing experiments.
2.  In terms of graphics, we delivered a first version using rudimentary figures (an l-shape, an e-shape, a square and an angle) with no correlation with the elements commonly used in the industry 4.0. Then, we modified the grammar version and created more representative figures such as a router, production line and two types of walls (vertical and horizontal).
3.  In terms of logic, we improved our fitness function by first penalizing big objects obstructing other figures (greater than 500 px) and then penalizing overlapping figures. We were able to produce more meaningful and realistic distributions.
4.  Improved performance, generating scenarios with more semantic meaning. In the beginning, was hard to find a balance between grammar expressiveness and computational effort, then we developed a flexible solution efficient enough to deliver a consistent solution in a reasonable time and a software architecture to easily handle the interaction between the 2D graphics engine and the ML algorithm.

The evolutionary process delivers several novel types of configurations, including unseen configurations that can help when testing a controller algorithm. However, despite the work done, the fitness function still requires some fine-tuning to achieve the possibility to target some complexity (e.g., having the chance to define an easy, medium or hard maze configuration and target that configuration inside a scenario), as well as making solvable the scenario itself (e.g., how can we know if the scenario itself is solvable?).

In the context of the Confirm Smart Manufacturing research centre, a national SFI research centre headquartered at the University of Limerick, we developed several use cases that solve several problems occurring in the manufacturing industry, adding flexibility and configurability to traditional on-site activities. For example, we enabled the remote programming and configuration of a cobot arm (collaborative robot) [24] through an

intuitive web application that communicates with and sends commands to the robot through a Domain Specific Language based on the robot's native API. In [25], we established a novel way to develop and deploy IoT applications through Model-Driven Development approaches. Creating the respective Domain Specific Language, we used the DIME and Pyrus application development platforms to develop an application that connects a range of heterogeneous technologies, including sensors using the EdgeX middleware platform [26] and a data analytics pipeline. We built this way innovative Low-code applications without needing full coding expertise nor requiring a mastery of the underlying runtime platform technologies. This way of handling complex domains through Domain Specific Language was already practiced in [27], where we showed how to implement controllers for robots using an abstract MDD approach based on APIs turned into a DSL.

This is conceptually very similar to the approach taken in MazeGen, where the grammars are an abstract DSL of the "language" of elements in our scenarios, and where MazeGen and PyGEVO themselves can be transformed into a DSL for the definition and optimization of Mazes. This approach will enable in the future a uniform definition of robot behaviours, navigation tools, and obstacle landscapes by means of respective DSLs that co-exist within the same model driven, generative application development platforms.

In terms of scalability and hierarchy, we intend to pursue a feature-based approach similar to the successful approach developed originally by [28] in the context of Intelligent Network telecommunication applications.

In the context of our MazeGen, the next steps include filtering only valid scenarios, increasing the fitness score maintaining the semantic meaning of the mazes and finally also to avoid unnecessary collisions between the different figures. This work can also be extended to other kinds of robotic domains, such as the synthesis of robotic controllers for simulated environments [29] or also automatically synthesize collective behaviors for autonomous robots (evolutionary robotics) [30,31] but in the meantime, these steps represent a big challenge for the future work of MazeGen.

**Author Contributions:** Conceptualization, I.H.G. and T.M.; methodology, I.H.G. and T.M.; software, I.H.G.; validation, I.H.G. and T.M.; formal analysis, I.H.G.; investigation, I.H.G.; resources, I.H.G.; data curation, I.H.G.; writing—original draft preparation, I.H.G. and T.M.; writing—review and editing, I.H.G. and T.M.; visualization, I.H.G.; supervision, T.M.; project administration, T.M.; funding acquisition, T.M. All authors have read and agreed to the published version of the manuscript.

**Funding:** This project received funding from Science Foundation Ireland (SFI) under Grant Number 16/RC/3918 (CONFIRM Centre) and 2094-1 (Lero, the Software Research Centre).

**Data Availability Statement:** https://github.com/IvanHGuevara/PyGEVO and https://github.com/IvanHGuevara/Results-MDPI-MazeGen (accessed on 11 April 2023).

**Conflicts of Interest:** The authors declare no conflict of interest. The funders had no role in the design of the study; in the collection, analyses, or interpretation of data; in the writing of the manuscript, or in the decision to publish the results.

## Abbreviations

The following abbreviations are used in this manuscript:

| | |
|---|---|
| BNF | Backus-Naur Form |
| ML | Machine Learning |
| GE | Grammatical Evolution |
| NT | Non-Terminals |
| SIB | Service-Independent Building Block |

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
