# Peer review of "MazeGen: A Low-Code Framework for Bootstrapping Robotic Navigation Scenarios for Smart Manufacturing Contexts"

_electronics, doi:10.3390/electronics12092058_

Round 1

Reviewer 1 Report

The paper presents the MazeGen framework navigation scenarios using Grammatical Evolution for robots or drones for robotic navigation scenarios.

the outcomes of the experimentation show a low correlation between complexity and the number of obstacles placed and also provide evidence  of the emerging technology status.

The weaknesses of this work are related to the validation status, while authors present a wide experimentation set, the result validation still obscure and without benchmarking.

The authors pretend to change the grammar and the fitness function, still nor real results were provided 

the validation and experimentation part as well as the methodology need much more elaboration. The article is too short and require to be detailed and enriched. 

Author Response

Response to Reviewer 1 comments

Point 1: The weaknesses of this work are related to the validation status, while authors present a wide experimentation set, the result validation still obscure and without benchmarking. The validation and experimentation part as well as the methodology need much more elaboration. The article is too short and require to be detailed and enriched. 

Answer: As requested, more details regarding performance/fitness scores were added to the paper. All the details were uploaded to GitHub: https://github.com/IvanHGuevara/Results-MDPI-MazeGen

Reviewer 2 Report

 In this research, the authors described the MazeGen framework (as maze generator), which generates navigation scenarios using Grammatical Evolution for robots or drones to navigate. The maze generator uses evolutionary algorithms to create robotic navigation scenarios of different complexity  levels along a scenario profile. Grammatical Evolution is a Machine Learning technique from the  Evolutionary Computing branch that uses a BNF grammar to describe the language of the possible scenario universe and a numerical encoding of individual scenarios along that grammar. Through  a mapping process it converts new numerical individuals obtained by operations on the parents’ encodings to a new solution by means of the grammar. In this context, the grammar describes the  scenario elements and some composition rules. The authors also analyse associated concepts of complexity,  such as cost of production of the scenario or skill levels needed to move around. Preliminary results show a low correlation between complexity and the number of obstacles placed, as complex configurations were found in early stages of the evolution process. They also evidence the emerging technology status of this approach, as we still need to find out how to reliably find solvable scenarios and characterize those belonging to the same class of equivalence. Despite being an emerging technology, MazeGen allows users to simplify the process of building configurations for  smart manufacturing environments, by making it faster, more efficient and reproducible, and it also puts the non-expert programmer in the center of the development process, as little boilerplate code is needed. Generally, this is a good work. It can be accepted if the authors can consider the following issues:1. The motivation of the work should be well organized in the introduction part. 2. How to define the low-code framework? 3. Is the proposed method applicable for other kinds of robotics? 4. More related works for robotics are welcome to enrich the literature review such as Tuning of Digital PID Controllers Using Particle Swarm Optimization Algorithm for a CAN-Based DC Motor Subject to Stochastic Delays;Fault Diagnosis of an Autonomous Vehicle With an Improved SVM Algorithm Subject to Unbalanced Datasets. 5. Please double check the language of the work. 

Author Response

Comments to the Reviewer 2:

Point 1: The motivation of the work should be well organized in the introduction part. 

Answer: Motivation was added in the introduction to further clarify our intentions.

Point 2: How to define the low-code framework?

Answer: We added more information about our framework and detailed more about our results.

Point 3: Is the proposed method applicable for other kinds of robotics? 

Answer: We added more work done about evolutionary robotics and swarm robotics (conclusions) that could be done with our solution.

Point 4: More related works for robotics are welcome to enrich the literature review such as Tuning of Digital PID Controllers Using Particle Swarm Optimization Algorithm for a CAN-Based DC Motor Subject to Stochastic Delays; Fault Diagnosis of an Autonomous Vehicle With an Improved SVM Algorithm Subject to Unbalanced Datasets

Answer: Thanks for these recommendations. They are very informative for us.

Point 5: Please double check the language of the work.

Answer: Done

Reviewer 3 Report

1) The introduction should be improved with more recent applications about AI techniques in the industrial cases such as deep learning, reforcement learning, Gaussian process regression etc. Because your research work is based on data-driven technique. Please refer to 10.1109/TTE.2022.3209629. 

2) The figures should be reproduced. They are difficult to understand for the readers. In addition, more schematique explanations about your studied system, the used AI techniques should be given and explained in detail. 

3) More statistical information should be given in the abstract and conclusion parts.  

Author Response

Comments for Reviewer 3:

Point 1:  The introduction should be improved with more recent applications about AI techniques in the industrial cases such as deep learning, reforcement learning, Gaussian process regression etc. Because your research work is based on data-driven technique. Please refer to 10.1109/TTE.2022.3209629. 

Answer: Thanks for this recommendation, it is very informative for us.

Point 2: The figures should be reproduced. They are difficult to understand for the readers. In addition, more schematique explanations about your studied system, the used AI techniques should be given and explained in detail. 

Answer: More information for about our figures and their meaning was added in the experimentation section.

Point 3: More statistical information should be given in the abstract and conclusion parts.  

Answer: We added more metrics in the experimentation section to give a better understanding.

Reviewer 4 Report

Well-presented research with scientific soundness, although some issues need to be clear before further process. 

There are some grammar errors and typos like in line 26: "...such a a way.."

The related work is missing more studies, the generation of the contribution of the research, as well as motivation for this research.

The discussion of the results should be deeper including a comparison with other studies by other authors.

The authors should present the conclusions more specifically and quantify them (in the form of bullets).

Author Response

Comments for Reviewer 4

Point 1:  There are some grammar errors and typos like in line 26: "...such a a way.."

Answer: The paper was rechecked and reviewed in terms of grammar.

Point 2: The related work is missing more studies, the generation of the contribution of the research, as well as motivation for this research.

Answer: More use cases were added to the paper, also more work related to our approach.

Point 3: The discussion of the results should be deeper including a comparison with other studies by other authors.

Answer: More studies (evolutionary robotics and swarm robotics) were added to compare with ours

Point 4: The authors should present the conclusions more specifically and quantify them (in the form of bullets).

Answer: Conclusiones were presented in a more straightforward way

Round 2

Reviewer 1 Report

the abstract wasn t updated. the result should be reflected

Auhtors have done several updates. The following point should be considered:

please edit equation in Figure 19 abbd19 using appropriate equation tool

Is there any limitation about the shape type, please explain within the methodology

when the shape is complex, what do you suggest

the 

Author Response

Responses to Reviewer 1:

Point 1: the abstract wasn t updated. the result should be reflected

Answer: The abstract was updated, addressing the concerns about complexity and adding information about our new statistics.

Point 2: please edit equation in Figure 19  using appropriate equation tool

Answer: Done with pure LaTeX

Point 3: Is there any limitation about the shape type

Answer: A new subsection called "Choosing our figures" addresses that question

Point 4:  please explain within the methodology when the shape is complex, what do you suggest

Answer: Thanks for mentioning this. We changed the way of addressing the word "complex" or "complexity".  We defined "complexity" in the abstract as the way of moving around the maze and the cost of production of the scenario. The following sentences were changed to have more precise definitions:

Line 235: Making the experiment more costly in terms of computing resources

Line 424: Improved performance, generating scenarios with more semantic meaning

Line 10: We also analyze associated concepts of complexity, understanding complexity as the cost of production of the scenario and skill levels needed to move around the maze

Line 13: Preliminary results show a low correlation between complexity and the number of obstacles placed, as configurations with more difficult obstacle dispositions were found in the early stages of the evolution process

Line 25: Robotic Navigation Scenarios are configurations with obstacles and objects, like walls, tables, machines, production lines,  etc, in a typically delimited or enclosed space, that enable us to test different types of requirements over a spatial model.

Line 348: The initial grammar had a triangle, a square, an e-shape, and an l-shape as basic forms, and included recursion to allow scenarios with possibly infinite recursive figures. 

Line 432:  However, despite the work done, the fitness function still requires some fine-tuning to achieve the possibility to target some complexity (e.g., having the chance to define an easy, medium or hard maze configuration and target that configuration inside a scenario)

Reviewer 3 Report

The questions have been answered the paper can be accepted

Author Response

Thanks for your valuables contributions

Reviewer 4 Report

No further comments.

Author Response

Thanks for your valuables contributions